# The implications of APOBEC3-mediated C-to-U RNA editing for human disease
Melissa Van Norden [1], Zackary Falls [1], Sapan Mandloi[1], Brahm H. Segal [1,2], Bora E. Baysal[2], Ram Samudrala [1] & Peter L. Elkin [1,3,4] ✉

Intra-organism biodiversity is thought to arise from epigenetic modification of constituent genes and post-translational modifications of translated proteins. Here, we show that post-transcriptional modifications, like RNA editing, may also contribute. RNA editing enzymes APOBEC3A and APOBEC3G catalyze the deamination of cytosine to uracil. RNAsee (RNA site editing evaluation) is a computational tool developed to predict the cytosines edited by these enzymes. We find that 4.5% of non-synonymous DNA single nucleotide polymorphisms that result in cytosine to uracil changes in RNA are probable sites for APOBEC3A/G RNA editing; the variant proteins created by such polymorphisms may also result from transient RNA editing. These polymorphisms are associated with over 20% of Medical Subject Headings across ten categories of disease, including nutritional and metabolic, neoplastic, cardiovascular, and nervous system diseases. Because RNA editing is transient and not organism-wide, future work is necessary to confirm the extent and effects of such editing in humans.

Our biodiversity has long been thought to come from alternative splicing and post-translational modification of the thousands of proteins encoded in the human genome. There are about 20,300 protein-encoding genes in the genome, of which 19,267 have an approved HUGO gene name (as of 04/07/23)[1,2]. However, about 70,000 proteins result from splice variants, and thousands more could result from post-translational modifications[2]. The expansion of proteoforms from genes is multifactorial, we hypothesize, and it could be due to mechanisms beyond post-translational modifications. In this paper, we explore the ability of RNA editing, a post-transcriptional modification, to contribute to human biodiversity.

RNA editing has been described as "any site-specific alteration in an RNA sequence that could have been copied from the template, excluding changes due to processes such as RNA splicing and polyadenylation."[3] There are two well-studied families of RNA editing enzymes that catalyze single nucleotide substitutions[4]. The ADAR (adenosine deaminase acting on RNA) family deaminates adenosine to guanosine-analog inosine (A > I) in specific double-stranded RNA contexts[5]. Similarly, specific APOBEC (apolipoprotein B mRNA editing catalytic polypeptide-like) family enzymes deaminate cytosine to uracil (C > U) in single-stranded contexts[6,7].

Two such APOBEC-family enzymes are APOBEC3A and APOBEC3G. The ability of APOBEC3 enzymes to edit RNA is a relatively recent discovery, starting in 2015 with a paper by ref. 8. For a long time, these enzymes were primarily known for their ability to edit single-stranded DNA (ssDNA) produced by viruses such as HIV (APOBEC3G) and parvovirus (APOBEC3A)[6]. Perhaps because of this, known APOBEC3-mediated RNA editing sites are rare. In a dataset taken from a paper by Asaoka et al., only about 0.05% of bases across 2343 genes were considered APOBEC3A/G RNA editing sites[9]. We contend that, beyond their effects on viruses, these enzymes may also affect human health via the editing of healthy human mRNA and the subsequent creation of protein variants.

Previous work has shown that APOBEC3A/G cytosine deaminases preferentially target RNA and ssDNA substrates with stem-loop structures[7,10]. Specifically, an optimal target contains a tri- or tetraloop, with the edited cytosine at the 3' end of the loop and a pyrimidine 5' to it (Fig. 1a)[7,10]. Specific cytosines for which APOBEC3 enzymes have high affinity, such as c.136 in SDHB, have previously been observed to undergo RNA editing even in normal physiological circumstances[8,11]. Additionally, APOBEC3-mediated RNA editing activity is known to transiently increase in monocytes, peripheral blood cells, and blood-brain barrier cells upon environmental interferon exposure[8,12–15]. A combination of environmental factors and a high-affinity site can lead to high levels of editing; editing was observed in over half of SDHB transcripts in megakaryocyte-erythroid progenitor (MEP) cells exposed to interferon-1 and hypoxic conditions[8]. Thousands of cytosines are known targets of APOBEC3-mediated editing,

[1]Department of Biomedical Informatics, Jacobs School of Medicine and Biomedical Sciences, University at Buffalo, State University of New York, Buffalo, NY, USA. [2]Roswell Park Comprehensive Cancer Center, Buffalo, NY, USA. [3]Department of Veterans Affairs, VA Western New York Healthcare System, Buffalo, NY, USA. [4]Faculty of Engineering, University of Southern Denmark, Odense, Denmark. ✉e-mail: elkinp@buffalo.edu

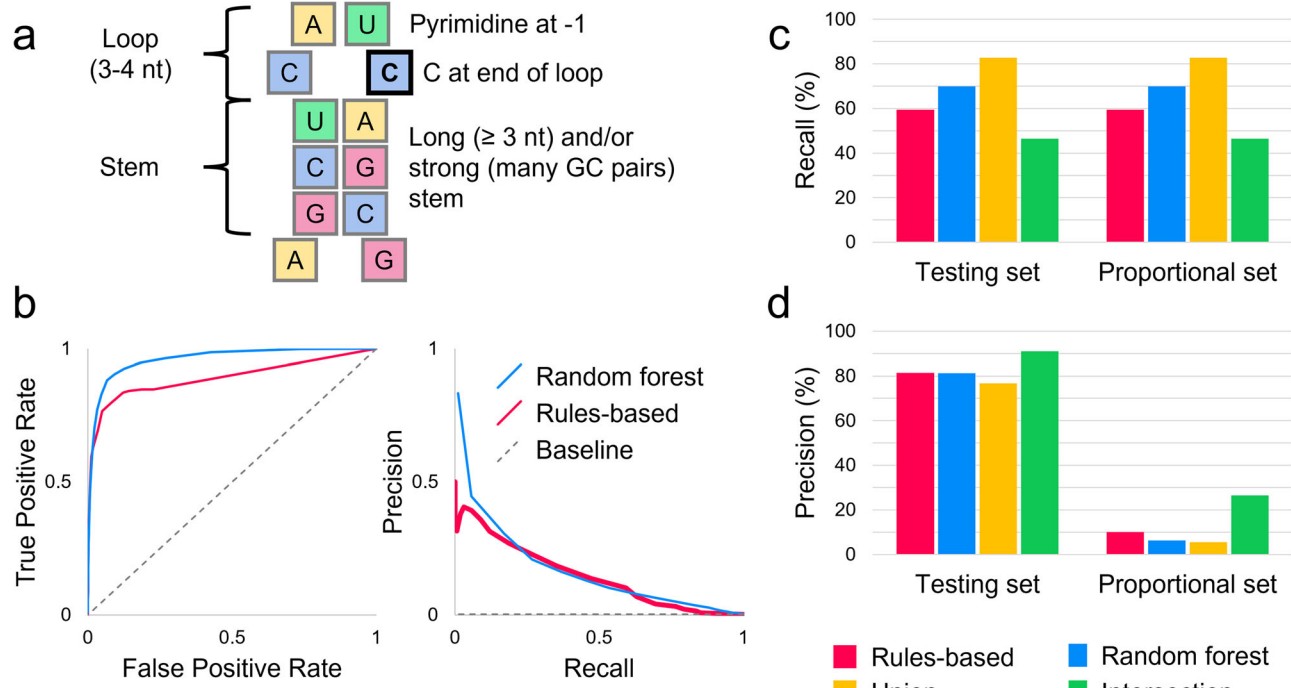

**Fig. 1 | Benchmarking results of RNAsee. a** APOBEC3A and APOBEC3G preferentially edit cytosines in stem-loop structures. The edited cytosine is outlined in black. In (**b–d**), red represents the rules-based, blue the random forest, yellow the union, and green the intersection model. (**b**) When tested on the proportional set, the random forest model had an AUROC of 0.962, and the rules-based model had an AUROC of 0.892. The random forest model also outperformed on average precision, with an AUPRC of 0.174 compared to the rules-based model's 0.147 and dataset baseline of 0.00213. The rules-based model's curve was similar to or higher than the random forest's at intermediate recall values, but lower at extreme values. The recall (**c**) and precision (**d**) of the two primary models and two consensus models were assessed on the testing set (editing:non-editing site ratio of 1:3) and the proportional set (editing:non-editing site ratio of 1:468). Because both sets contained the same positive sites, the recall was the same on both sets. The intersection model was the most precise on both sets, making it useful for selective studies; the union model had the greatest recall, so it can be used to survey potential editing sites more broadly.

editing at many of which result in non-synonymous or nonsense mutations[8,9,12]. Therefore, variant proteins that impact human health could be created by APOBEC3-mediated RNA editing at high-affinity sites and at an increased rate during and following tissue inflammation.

There is a growing body of evidence supporting the influence of C > U RNA editing on human health. APOBEC3-mediated RNA editing sites have been found in genes with known relationships to neoplasms, hypertension, and nervous system disorders, including amyotrophic lateral sclerosis (ALS), Alzheimer's, Huntington's, and Parkinson's disease[8,12]. Correlation between APOBEC3-mediated C > U RNA editing and diseases like epilepsy and sporadic Creutzfeldt-Jakobs disease (sCJD) has been found in mice, and a link between RNA editing and autoantigen creation in autoimmune diseases like systemic lupus erythematosus (SLE) has been suggested in humans[16–18].

However, this area of research is still relatively young. Most articles and tools on RNA editing still focus on ADAR-mediated A > I editing. For instance, the publicly available RADAR database catalogs A > I RNA editing sites with manual annotations, and this tool has been used alongside data from The Cancer Genome Atlas to identify protein variants that may affect tumor cell viability and drug sensitivity[19,20]. Although at least one study attempted to collate known APOBEC3A/G editing sites, no comprehensive database like RADAR yet exists for C > U editing events, and studies on APOBEC3 editing tend to remain narrowly focused on one or two diseases at a time[9].

To address this gap in knowledge, we have developed RNAsee (RNA site editing evaluation), a program that combines machine learning and rules-based methods to predict APOBEC3A/G-mediated RNA editing sites in transcripts of human genes[21]. In this paper, we will compare RNAsee-predicted APOBEC3A/G-mediated C > U editing sites with publicly available data on human protein variants from the ClinVar database, particularly pathogenic variants, to help open the conversation on the extent to which C > U RNA editing may contribute to human disease.

## Results
### Performance of RNAsee

To assess the performance of RNAsee in predicting APOBEC3A/G editing sites, we first benchmarked our models on the testing set, which was split 7:3 from the training-testing data and contained a 1:3 ratio of editing to non-editing sites (see Methods: Performance benchmark for RNAsee). Using a score threshold of ≥10 for the rules-based model and a probability threshold of >0.5 for the random forest model, we calculated recall, precision, F1 score, and Matthew's correlation coefficient (MCC) metrics for the two primary and two consensus models. The recall of the models was highest for the union model (82.8%) and lowest for the intersection model (46.5%), whereas the reverse was observed with precision (90.9% for intersection, 76.8% for union). Recall and precision of the other models are shown in Fig. 1c, d. On this dataset, F1 scores were highest for the union model at 0.80, then random forest at 0.75, rules-based at 0.69, and intersection at 0.62. MCC scores showed the same pattern, being highest for union at 0.73, then random forest at 0.68, rules-based at 0.62, and intersection at 0.58.

Because APOBEC3-mediated editing sites are rare by our estimation, making up only about 0.2% of the cytosines in the genes from our dataset, we were concerned that these results would be overly optimistic for a real prediction scenario. Therefore, we also benchmarked our models on a proportional set containing a 1:468 ratio of editing to non-editing sites (see Methods: Performance benchmark for RNAsee). Because no editing sites were included in this set that were not in the previous set, the recall was the same as on the testing set. Precision changed substantially, with the union's precision lowering to 5.5% and the intersection model's falling to 26.5%; however, these results still represent over 20x and 100x enrichments of

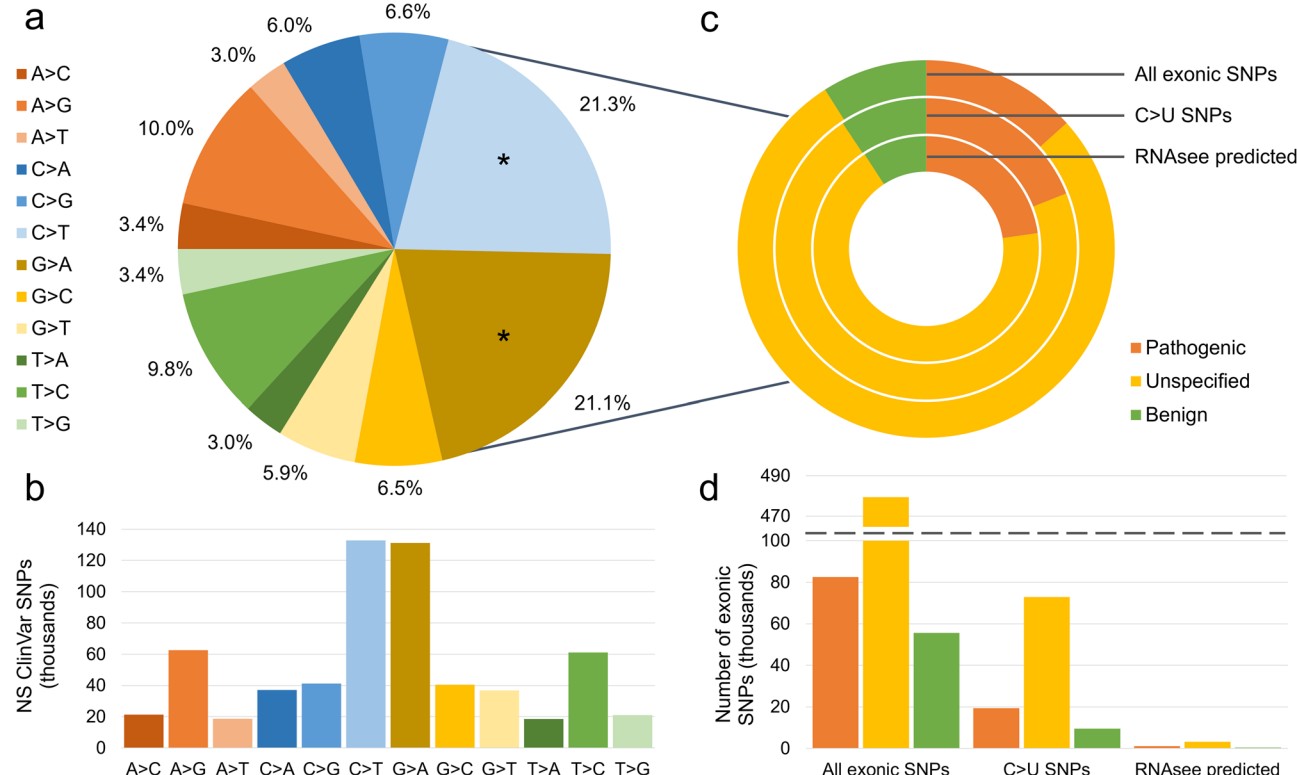

**Fig. 2 | Over two-fifths of non-synonymous SNPs in the ClinVar database may result from C > U DNA editing.** In (**a**, **b**), color represents the nucleotide change represented. **a** The recorded nucleotide changes of all non-synonymous (NS) single nucleotide polymorphisms (SNPs) in the ClinVar database were counted. Like-to-like or non-specific changes are excluded. Cytosine (C) to thymine (T) (21.3%) and guanine (G) to adenine (A) (21.1%) mutations may be associated with C to uracil (U) mutations when transcribed into RNA, depending on which strand is the template; the slices representing these changes are asterisked. The raw number of each type of NS SNP is plotted in (**b**). In (**c**, **d**), orange represents pathogenic SNPs, green benign, and yellow those of unspecified pathogenicity.

**c** Each ClinVar SNP is also associated with a pathogenicity label. These were sorted into three bins: pathogenic, unspecified, and benign. The proportions of each label among all exonic SNPs, the subset of exonic SNPs associated with C > U RNA changes, and all SNPs returned as possible APOBEC3 editing sites by RNAsee are shown. The raw number of SNPs per set are shown in (**d**). 22.7% of the potential editing sites returned by RNAsee were labeled as likely pathogenic or pathogenic, whereas only 9.2% were labeled as likely benign or benign. This suggests that C > U RNA editing has a substantial possibility of negatively influencing human health.

positive sites relative to the baseline, respectively. These results are also shown in Fig. 1c, d. F1 and MCC scores similarly fell: F1 scores were 0.34 for intersection, 0.17 for rules-based, 0.12 for random forest, and 0.10 for union, and MCC scores were 0.35 for intersection, 0.24 for rules-based, and 0.21 for random forest and union.

Finally, for a more general assessment of our primary models, the area under the receiver-operator characteristic (AUROC) and the area under the precision-recall curve (AUPRC) metrics were calculated on the proportional set (Fig. 1b). Because the consensus models have two thresholds (one corresponding to each primary model), they were not included in these assessments. The AUROC of the random forest model was 0.962, which is higher than the rules-based model's AUROC of 0.892. However, the two curves overlapped at the highest thresholds (lowest false positive rates), including the points at which the thresholds for the rules-based and random forest models were set; the separation at lower thresholds/higher false positive rates is likely due to the rules-based model's strict exclusion criteria, which limits its maximum recall (see Methods: RNAsee). The AUPRC of the random forest model was also higher at 0.174 compared to the rules-based model's 0.147 and the baseline AUPRC of 0.00213.

### Identification of possible editing sites

To start, we needed a set of nucleotide polymorphisms that result in protein variants. We extracted an annotated set of DNA mutations from the ClinVar database[22]. We pared this set down to only include single nucleotide polymorphisms (SNPs) that were exonic and non-synonymous. Then, we examined the relative frequency of different polymorphisms (Fig. 2a, b).

Among 622,622 non-synonymous SNPs, C > T (21.3%) and G > A (21.1%) were by far the most frequently observed polymorphisms. Both of these types of SNPs may result in C > U changes in RNA, depending on which DNA strand is transcribed. Therefore, to determine which of these SNPs could correspond with C > U editing, we needed to refer to the RNA sequences.

We examined coding sequence files for each gene with an SNP. Entries in genes with no good coding sequence file found were excluded, leaving 617,363 SNPs across 9228 genes. This set is the set of all exonic SNPs. Of these, 101,565 (16.5%) were associated with a C > U RNA change and were included in the set of C > U SNPs. We used the most sensitive RNAsee model, the union model, to analyze each cytosine in that set. RNAsee returned 4600 SNPs, 4.5% of the C > U SNPs, as potential APOBEC3A/G-mediated RNA editing sites. These SNPs were included in the RNAsee predicted set. Of these 4600 sites, 62 (1.34%) are known APOBEC3-mediated RNA editing sites included in the Asaoka et al. set[9].

Each entry was annotated with a pathogenicity tag, which we sorted into three bins: pathogenic, benign, and unspecified. The proportions of these tags in the sets of all exonic SNPs, C > U SNPs, and RNAsee predicted sites were calculated (Fig. 2c, d). Of the changes in the C > U SNPs set, 19,285 (19.0%) were tagged as pathogenic, 9459 (9.3%) were tagged as benign, and 72,821 (71.7%) were unspecified. The percentage of pathogenic SNPs were higher in the RNAsee predicted set; in this set, 1046 (22.7%) of SNPs were tagged as pathogenic, 3132 (68.1%) as unspecified, and 422 (9.2%) as benign. This also meant that a higher percentage of pathogenic C > U SNPs were

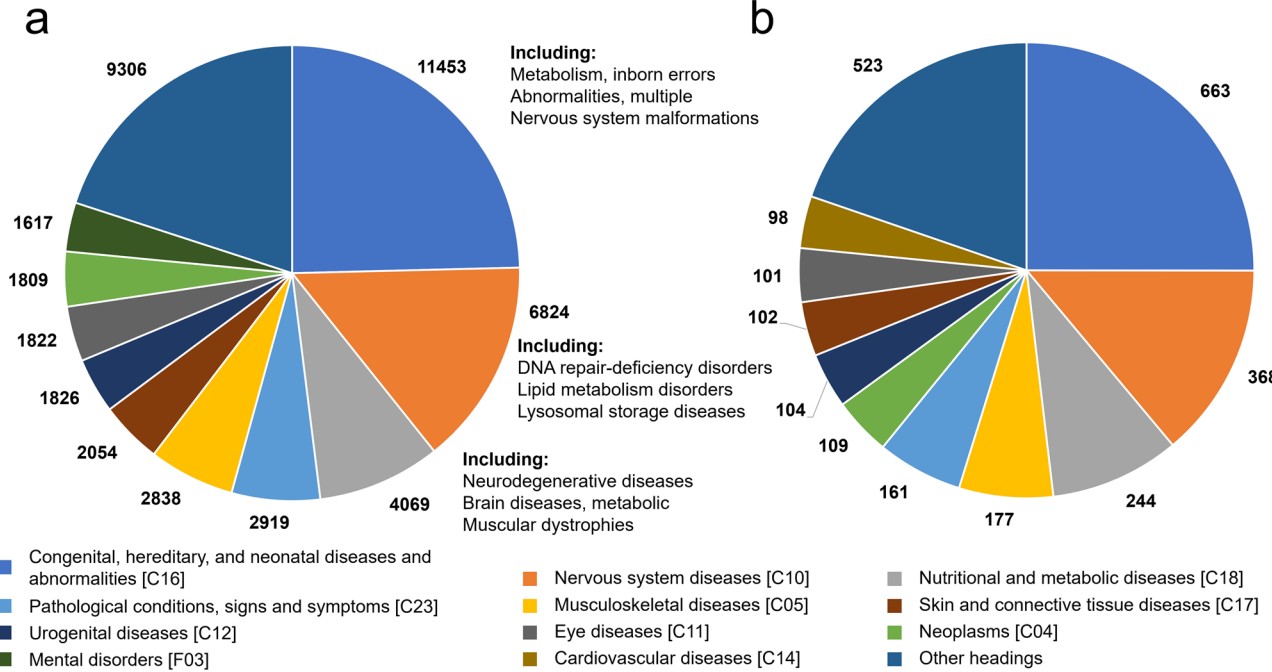

**Fig. 3 | Potential pathogenic APOBEC3A/G targets are associated with similar diseases to the set of all C > U SNPs.** The number of times the top ten most common top-level Medical Subject Headings (MeSH) are associated with the set of pathogenic C > U SNPs (**a**) and the pathogenic RNAsee predicted set (**b**) are shown, with each colored slice representing a different subject heading.

The most common types of conditions are similar between the two groups, with congenital abnormalities, nervous system diseases, and nutritional and metabolic diseases being the most common types of conditions associated with both sets.

predicted editing sites (5.4%) than unspecified (4.3%) or benign (4.5%). The percentage of benign SNPs was relatively consistent between the sets, but the percentage of pathogenic sites was highest in the RNAsee predicted set and lowest in the set of all exonic SNPs.

## Associations with disease

Of 101,565 sites in the C > U SNPs set, 72917 (71.8%) were associated with at least one disease, condition, or phenotype in ClinVar. To determine which areas of health are most likely to be affected by APOBEC3A/G-mediated RNA editing, we decided to survey these conditions.

We labeled each SNP with the Medical Subject Headings (MeSH) that most closely matched its associated conditions. In total, there were 6534 unique conditions associated with C > U SNPs. 575 conditions (8.8%) could not be matched with a good single MeSH equivalent and were labeled "Not found." In total, 7181 SNPs (9.3%) were associated with "Not found" at least once, including 401 SNPs considered pathogenic.

The top-level subject headings corresponding to the initial MeSH labels were found for each SNP, with each top-level term labeling an SNP no more than once. We counted the number of times each top-level subject heading or "Not found" was associated with an SNP in the C > U SNPs and RNAsee predicted sets. Because this work is focused on the potential effects of RNA editing on human health, we focused on those SNPs labeled pathogenic. The top ten most common top-level MeSH subject headings associated with pathogenic SNPs in the C > U SNPs and RNAsee predicted sets are shown in Fig. 3a, b, respectively.

In both sets, the most common associated heading was congenital, hereditary, and neonatal diseases and abnormalities. Because ClinVar pathogenic variations are thought to be causally linked to their associated conditions, all conditions associated with these variations should either be congenital or result from a de novo tissue mutation, for instance, in a neoplasm. Nervous System Diseases and Nutritional and Metabolic Diseases were the second and third most common subject headings in both groups, which may mean C > U RNA editing is most likely to negatively influence human health in these two areas. The proportions of most subject

headings were similar between the two groups, with only slight differences in their relative rankings.

To further elucidate the potential effects of APOBEC3-mediated RNA editing on human health, we wanted to examine the proportion of diseases with any possible influence from APOBEC3-mediated RNA editing. To do this, we counted the number of third-level MeSH headings (grandchildren of top-level headings) with at least one predicted editing site. To find an upper bound on the number of conditions affected, we also counted the number associated with at least one C > U SNP (Fig. 4).

Almost all top-level types of non-traumatic, non-infectious conditions recognized by MeSH had at least one grandchild term associated with at least one predicted editing site. Some types of diseases had as few as 4% of their grandchild terms with an associated editing site (otolaryngologic diseases, respiratory tract diseases), whereas 72% of third-level nutritional and metabolic diseases were associated with predicted editing sites. Notably, some top-level headings with a large number of predicted editing sites had a lower percentage of diseases with associated predicted editing sites.

## Discussion

We searched a list of known non-synonymous DNA SNPs to see which of those polymorphisms, and the resultant variant proteins, could also be caused by APOBEC3A/G-mediated RNA editing. We found that about 4.5% of known SNPs that result in C > U mRNA changes are also potential APOBEC3A/G editing sites. If APOBEC3A/G RNA editing regularly occurs at even a fraction of these sites, this could result in meaningful effects on human health. It has previously been demonstrated that interferon-rich environments, such as inflamed tissues, increase APOBEC3-mediated RNA editing in a variety of cell types[8,12–15]. Transient increases in RNA editing could result in the production of variant proteins, which could affect recovery time and result in sequelae following periods of inflammation.

We only examined those SNPs in ClinVar that are transcribed into RNA and, ultimately, translated into proteins. The effects on human health associated with these SNPs should, therefore, largely result from the creation of variant RNA and proteins. RNA editing can result in over 50% transcript

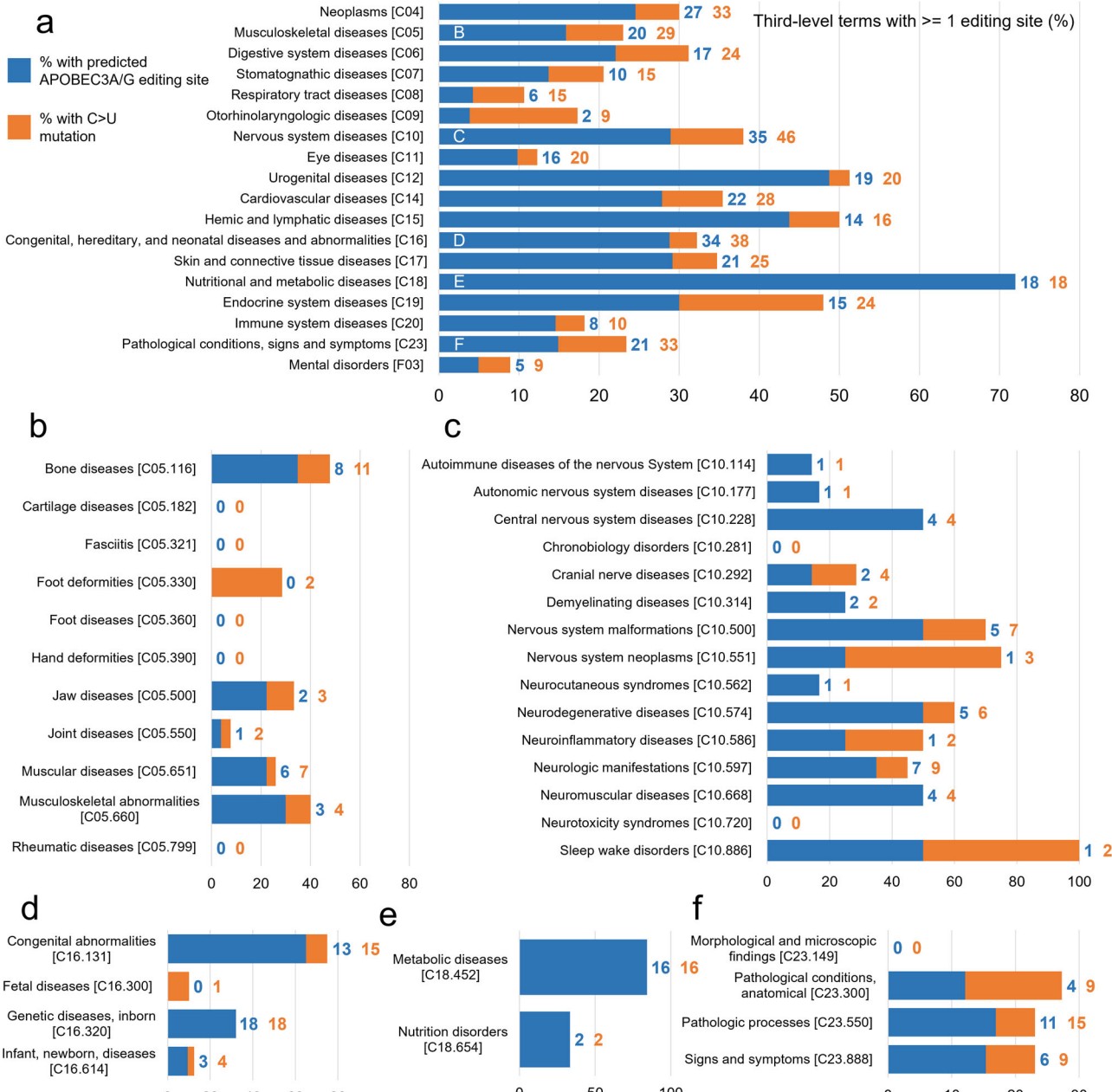

**Fig. 4 | Proportion of conditions associated with at least one potential APOBEC-mediated RNA editing site. a** For select top-level MeSH terms, the number of grandchild terms with at least one predicted APOBEC3A/G editing site was calculated (blue). In addition, the number of grandchild terms with at least one associated C > U mutation, but no predicted editing site, was found (orange). This was divided by the total number of grandchild terms for each top-level term to approximate the percentage of diseases associated with each top-level term that could be influenced by C > U RNA editing at a known site of genetic variation. The total number of terms counted can be found at the end of each bar, and the tree number of each term can be found in brackets at the end of its label. Additional graphs, showing the percent of child terms for each second-level term, are provided for the child terms of the top five most common top-level terms: **b** Musculoskeletal diseases, **c** nervous system diseases, **d** congenital, hereditary, and neonatal diseases and abnormalities, **e** nutritional and metabolic diseases, and **f** pathological conditions, signs and symptoms. C > U RNA editing may play a role in at least one disease for almost every category of non-acquired human disease.

alteration when a high-affinity sequence and activating environmental factors are involved[8]. We contend, therefore, that transient instances of high editing activity could cause the transient creation of these same variant RNA and proteins. This, in turn, could result in similar dysfunctions on the cellular or tissue level as are observed in the DNA SNPs recorded in ClinVar. Of course, these effects would be less universal and more transient, but they could still have serious consequences if they occurred in the wrong tissues, such as the brain or heart, created the wrong type of variant protein, such as plaque-forming proteins or autoantigens, or occurred alongside a

heterozygous DNA mutation affecting the same gene, altering the wild-type RNA and increasing the expression of a dosage-sensitive mutation such as *Pten* mutations[23].

Our results further support the validity of considering the sites we examined as potential RNA editing sites. Firstly, we primarily examined those 4600 sites returned by RNAsee as potential editing sites. If RNAsee performed in the application as it did in benchmarking, we would expect at least 250 (5.5%) of these sites to actually undergo RNA editing, resulting in the creation of variant transcripts and proteins (see Fig. 1d). Therefore, our

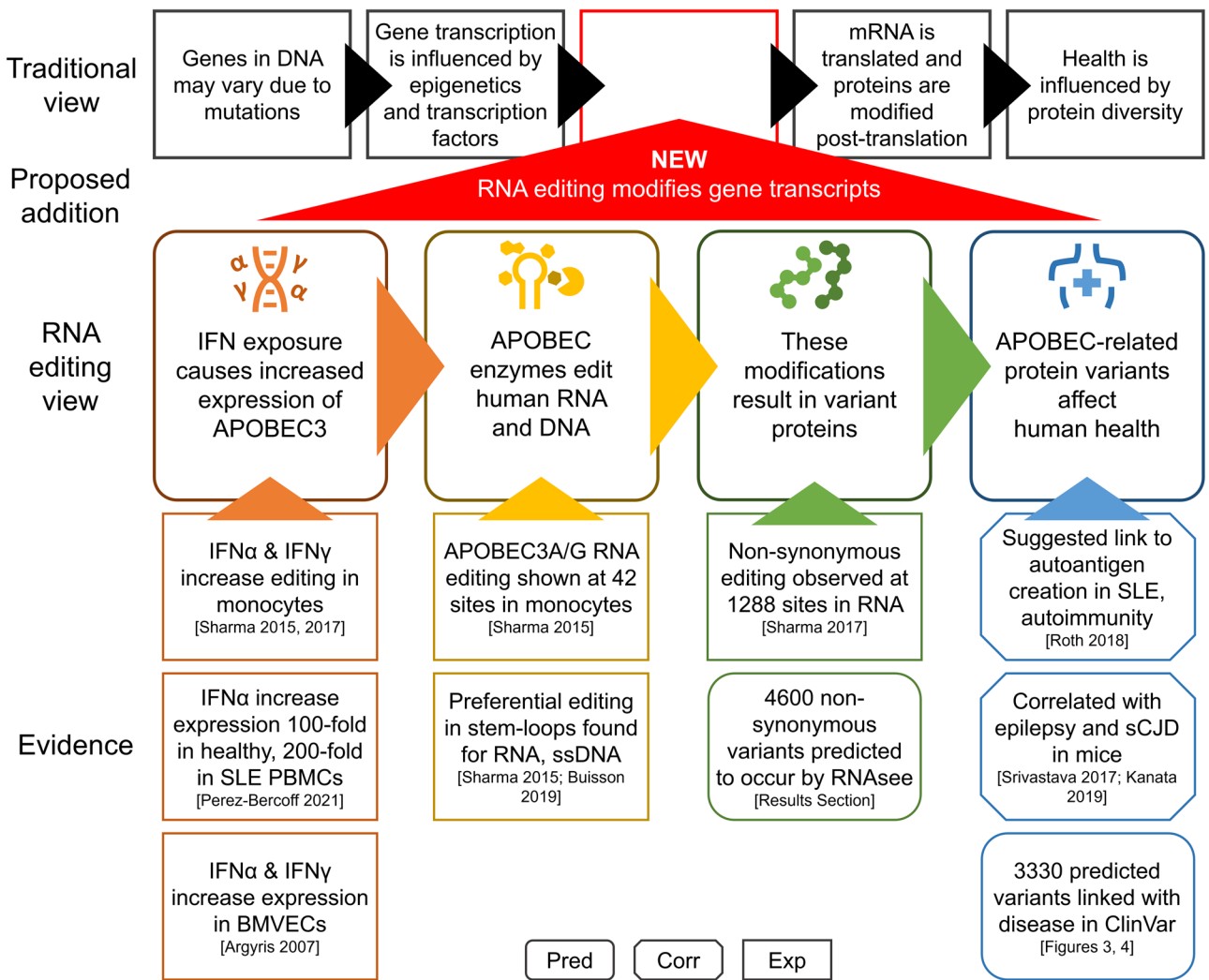

**Fig. 5 | Selected evidence for the influence of RNA editing on human health.** In the traditional model of biology, protein diversity results from mutations in DNA and post-translational modifications, and the expression of genes is regulated via epigenetics and transcription factors. We propose adding RNA editing to this view. Current evidence suggests that APOBEC-family enzymes are expressed when some cell types are exposed to environmental IFN. Some APOBEC enzymes, such as APOBEC3A and APOBEC3G, have been shown to edit human RNA. This editing has been shown to cause non-synonymous changes in mRNA, and we used RNAsee to predict additional sites that may also undergo editing. The effects of this editing on human health are still under-researched, but some prior work has suggested links between APOBEC-mediated RNA editing and certain autoimmune and neurological disease. Our predictions additionally suggest that at least one potential editing site is associated with most disease categories recognized in MeSH subject headings. It is essential that future work attempts to investigate and confirm or refute the links between RNA editing and human health as proposed in this and other works. *The box shape of evidence boxes represents the type of study the information is taken from. Rounded edges are computational (pred)ictions, cut corners are (corr) elational studies, and square corners are (exp)erimental studies. * Results 2: Identification of possible editing sites. PBMC: peripheral blood mononuclear cell; BMVEC: brain microvascular endothelial cells; IFN: interferon; SLE: systemic lupus erythematosus; sCJD: sporadic Creutzfeldt-Jakob disease.*

results support the overlap of RNA editing with DNA mutations. Secondly, our work suggests that some level of C > U deamination is likely contributing to the population of DNA SNPs recorded in ClinVar; over 40% of SNPs are associated with a C > T change on one strand, as opposed to less than 20% of SNPs associated with the other form of deamination, A > I/ A > G (see Fig. 2a). APOBEC3A is a major driver of C > U deamination in DNA, and previous works suggest that APOBEC3A edits RNA at similar sites as DNA[7,10,24]. Therefore, it is reasonable to conclude that, though the ClinVar SNPs were primarily found in DNA, they include a population of mutations that result from APOBEC3A DNA editing and are thus, when transcribed, likely to also undergo APOBEC3-mediated RNA editing.

When the union method of RNAsee was benchmarked, it returned about 3.2% of cytosines in the proportional set as potential APOBEC3A/G editing sites, including false positives. However, when run on C > U sites in the ClinVar database, it identified 4.5% (4600) of C > U SNPs and 5.4% (1046) of pathogenic C > U SNPs as potential editing sites. This increased

percentage supports the idea that some of the C > T and G > A polymorphisms recorded in ClinVar result from APOBEC3-mediated DNA editing, leading to a sample that was biased towards higher APOBEC3A/G editing affinity. APOBEC3A is already known to extensively edit DNA in neoplasms, but in this work, Neoplasms was only the ninth most common subject heading associated with SNPs that result in C > U changes in RNA and sixth most common subject heading associated with RNAsee-predicted editing sites (see Fig. 3)[10]. This suggests that C > U DNA editing likely factors into the creation of pathogenic SNPs and variant proteins beyond its known activity in neoplasms. APOBEC3A/G RNA editing at the same sites could likewise influence human health in varied areas.

We believe that these results, alongside existing evidence, support the significance of APOBEC3A/G RNA editing to human health (Fig. 5). Strong evidence suggests that expression of APOBEC3 enzymes increases in certain tissues when exposed to hypoxia or increased environmental interferons[8,12–15]. These conditions may be limited in area and duration, as

may be observed in acute viral infection; they may also be more widespread and long-lasting, as may be the case in chronic hypoxia resulting from emphysema or widespread inflammation resulting from autoimmunity. Such periods of increased enzyme expression provide increased opportunities for APOBEC3A/G-mediated mRNA editing events to occur, particularly in optimal stem-loop structures, and the effects on health may vary based on the location, duration, and intensity of the enzymatic activity[8,10].

These editing events have previously been shown to occur at locations that result in variant proteins due to missense or nonsense mutations[12]. Beyond these known sites, we have found that a much higher percentage of non-synonymous SNPs in the ClinVar database are associated with a C > T change on one DNA strand than is expected by random chance, which, we reason, indicates that C > U ssDNA editing has contributed greatly to the set of known human SNPs. Since similar substrates are targeted and similar sites are edited by APOBEC3A/G enzymes in DNA and RNA, this suggests more frequent RNA editing is likely occurring at some or all of the 16% of non-synonymous SNPs associated with C > U RNA changes[7,10,24]. RNAsee specifically suggested that 4600 of these sites, or 4.5% of all C > U SNPs, are particularly probable editing sites, and 62 of those sites are known editing sites included in the Asaoka et al. dataset[9].

The ultimate question is: does this editing actually affect human health? Our work suggests it does. Of those sites suggested as probable RNA editing sites by RNAsee, 1046 were tagged as pathogenic, and an additional 3132 have unknown or unspecified pathogenicity. Over 20% of third-level MeSH subject headings associated with nutritional and metabolic, neoplastic, cardiovascular, and nervous system diseases (along with 6 other categories of disease) were associated with at least one predicted editing site. In addition, more studies are finding links between APOBEC-mediated RNA editing and human disease. For instance, APOBEC-mediated editing has been correlated with epilepsy and sCJD in mouse models[16,17]. One study on the linkage between SLE and RNA editing even suggested RNA editing as a mechanism for autoantigen formation in autoimmune diseases[18]. Therefore, there is a high likelihood that RNA editing affects human health in some or all of the areas of disease noted in this study, particularly when there is an environmental pressure causing increased APOBEC3 activity.

This paper uses the effects on the health of DNA SNPs to infer the effects of RNA editing. However, as noted, RNA editing is a more transient, less universal process than DNA mutation, and, in real life, its effects may vary based on the enzyme's activity, the duration of this activity, and the tissue affected. On the other hand, this paper only considers the activity of the APOBEC3A/G enzymes. If APOBEC3A and APOBEC3G could cause polymorphisms at some of the loci identified in this paper and mediate some of the disorders noted, it seems likely that other APOBEC or ADAR-family enzymes are simultaneously editing RNA at entirely distinct sites, activated by distinct conditions, and with distinct effects on health. Therefore, to fully explore this mechanism behind protein diversity, future research should be devoted to (1) describing the sites at which RNA editing occurs, (2) finding the conditions in which these enzymes are most active, and (3) identifying the effects of variant RNA or proteins resulting from RNA editing on protein diversity and human health. In this way, we can add to our basic understanding of human molecular biology.

Classically, our biodiversity is thought to come from our constitutive genetics, epigenetic phenomenon, transcriptional differences, and post-translational modification of proteins. Here, we have shown evidence that RNA editing could also play a role in creating the variant proteins that contribute to human disease. Previous works have shown that the extent of RNA editing is sensitive to environmental factors such as interferon presence and hypoxia, and, in an era where worries about our changing environment are ever- increasing, understanding how environmentally sensitive mechanisms like RNA editing affect our cells is essential[8,12,14,15]. Future research will apply our analysis to the transcriptomes and proteomes of specific disease conditions in order to further clarify the functional and predictive role of APOBEC-mediated C > U RNA editing in human disease.

## Methods

### Extraction of disease variant data

The ClinVar database, maintained by the National Center for Biotechnology Information (NCBI), collects information regarding DNA variants found in patient samples, including assertions of clinical significance and evidence to support those assertions[22]. We extracted the annotated set of DNA mutations from ClinVar on May 18, 2022, via FTP[25]. All variants that were not SNPs (Type of single nucleotide variant) were excluded. Two records were provided for each variant corresponding to the GRCh37 and GRCh38 reference genomes. Records corresponding to GRCh37 were excluded. Finally, because this study was intended to examine RNA editing as a potential source of protein diversity in human cells, all DNA variants that would not result in protein variants (including non-exonic or exonic but synonymous variants) were excluded.

For each SNP, we extracted the following information: the gene, allele ID, any amino acid change resulting from the mutation, assertion of clinical significance, rsID, original and mutant nucleotides, and a list of phenotypes associated with the variant. To get a baseline for the frequency of different SNPs, we used the original and mutant nucleotides (found in the ReferenceAlleleVCF and AlternateAlleleVCF columns) for each SNP. Entries with non-specific nucleotides (e.g., A > X) or like-to-like nucleotide changes (e.g., A > A) were excluded. Entries without any associated amino acid change recorded were also excluded. We also binned the clinical significance into three categories (pathogenic, benign, or unspecified) based on whether the significance tag contained the words pathogenic, benign, or neither.

Coding sequence files were obtained from the consensus coding sequence (CCDS) database[26]. An attempt was made to download a human coding sequence file for any gene with at least one SNP that matched the inclusion criteria. If the said file did not exist or was incompatible with the data contained within ClinVar (i.e., the original amino acid in the record is not coded for by the codon at the corresponding location in the coding sequence file), variants in that gene were excluded. Out of 622,222 SNPs matching the inclusion criteria, 617,363 exonic single nucleotide polymorphisms across 9228 genes were analyzed. The amino acid change recorded in each entry and the corresponding codon in the coding sequence file were used to determine whether each SNP could be associated with C > U RNA editing. Those DNA polymorphisms that resulted in a C > U change in the transcribed mRNA were included in the C > U set and analyzed using RNAsee. In total, 101,565 sites were assessed for their likelihood of being APOBEC3A/G editing sites.

### RNAsee

RNAsee is a publicly available Python package. It can be found at https://github.com/ram-compbio/RNAsee. We previously developed RNAsee to identify potential APOBEC3A/G editing sites. RNAsee v1 was a rules-based method that used a simple stem-identification algorithm and a minimum free energy score assigned by ViennaRNA to rank cytosines within a gene from most to least likely to undergo RNA editing [21,27]. During the development of RNAsee v2, we decided to utilize the set of known APOBEC3-mediated RNA editing sites published in Asaoka et al. to both improve and benchmark RNAsee[9]. All cytosines identified by Asaoka et al. were considered editing sites, and all other cytosines in the same genes were considered non-editing sites[9].

We refined the rules-based model to score sites based on sequential features commonly observed in this dataset in addition to the presence and strength of a stem-loop structure. We developed a scoring algorithm, and we trained a scoring threshold for the rules-based model for best F1 score on a subset of the dataset.

We also added a machine-learning model to RNAsee. Four types of classification models were initially considered: support vector machines, logistic regression, decision tree, and random forest. We also considered different methods of vectorizing the sequence surrounding the potential editing site based on the number of nucleotides included (a 25- or 15-nucleotide stretch surrounding the site) and the method of encoding nucleotides (4 or 2 binary integers per nucleotide). Finally, we attempted to

accommodate the highly imbalanced class sizes between editing and non-editing sites (1:468) by downsampling non-editing sites and/or using SMOTE-based upsampling of editing sites. Ultimately, based on initial benchmarking performance and area under the receiver-operator curve (AUROC) figures, we chose a random forest model that takes as input a 25-nucleotide stretch surrounding a cytosine, vectorized into two binary integers per nucleotide (isPurine and pairsGC), trained on a set of sites downsampled for non-editing sites and with an increased proportion of non-editing sites that received high scores when scored by the rules-based model.

RNAsee v2 includes two primary methods and two consensus methods for editing site identification:

- **Rules-based model**. The rules-based model assigns a score to cytosines if, and only if, they are found at the 5' end of the loop of a stem-loop structure. The stem-loop structure is defined as a series of paired nucleotides surrounding a three to four nucleotide loop, with a single mismatch, or bulge, allowed two nucleotides 5' to the cytosine. Scores are then calculated based on the strength of the stem (3*(the number of GC pairs) + (the number of AU pairs)) and the presence or absence of specific sequential features (+2 for a uracil in the loop or a purine 5' to the cytosine and −2 for a guanine in the loop). If a cytosine receives a score of greater than nine according to these rules, it is considered an editing site.
- **Random forest model**. The random forest model was created as an instance of the RandomForestClassifier class from scikit-learn[28]. The model takes as input a 50-bit vector representing 15 nucleotides preceding and 10 following a cytosine. Every nucleotide is represented by 2 bits; one for whether that nucleotide is a purine and another for whether it can participate in GC pairing. The model was trained and tested using a 70–30 training/testing split of known editing and non-editing sites. A site is considered an editing site if its probability of being an editing site is over 0.5.
- **Consensus models**. The intersection and union models combine the outputs of the two primary methods using simple set operations. The intersection method outputs only sites returned by both primary models, and the union method outputs all sites returned by either primary model.

## Performance benchmark of RNAsee

Two datasets were used to benchmark RNAsee: the testing set and the proportional set. The former included all sites in the random forest model's testing set. It contained a 1:3 ratio of editing to non-editing sites. Because this ratio is greatly different from that found in the whole Asaoka et al. dataset, we also created the proportional set[9]. This set contains all sites found in the testing set plus additional non-editing sites to bring the ratio of editing to non-editing sites to 1:468, similar to the original ratio.

To ensure our benchmarking results would not be skewed by sequence redundancy in our positive and negative datasets, we assessed the sequence identity of each positive-positive and negative-negative pair on the 25 nucleotides used by the machine learning model around each cytosine (15 nucleotides before and 10 after the C). Less than 0.01% of pairings had over 75% sequence identity (at least 19 nucleotides in common) in both sets. In the positive dataset, about 50.6% of pairings had over 30% sequence identity, whereas 28.8% did in the negative dataset. This higher level of identity among positive sequences is expected, given they are all edited by APO-BEC3A/G enzymes, the activity of which is specific to certain sequential and structural features. Therefore, we do not expect that sequence redundancy is significantly biasing the model.

During benchmarking, each model was used to predict which cytosines in both datasets are APOBEC3 editing sites. The number of non-editing and editing sites in each model's predictions were tallied and used to calculate metrics including recall, precision, F1 score, and MCC. For the primary models, AUROC and AUPRC figures were also calculated.

Because this paper focused on fully exploring the potential effects of APOBEC3A/G-mediated RNA editing on human health, we decided to use

the union model, which showed the highest sensitivity (82.8%). For each SNP in the C > U set, the original cytosine in question and its sequential context were passed as input to the union model. All sites returned by the union model were considered potential APOBEC3A/G editing sites.

## Association with MeSH subject headings

The diseases and phenotypes associated with each site in the C > U set were extracted. All unique conditions were extracted into a single list. For each term in this list, the most relevant MeSH subject heading (descriptor or supplemental concept record) was found. The investigator was blinded as to which sites, predicted or otherwise, each term was associated with during this process. MeSH mappings were used to associate supplemental concept records with tree terms unless a mapping contained a qualifier that suggested the supplemental concept record was not a child of the mapped term (such as "abnormalities"). Each term was then associated with the chosen subject heading and every parent subject heading up to the top level. If no single subject heading encapsulated the given term, that term was associated with the phrase "Not found." "Not found" was treated as a top-level subject heading for analysis purposes. Finally, the terms were re-associated with the SNPs.

The number of times each subject heading appeared across all SNPs was counted. Each subject heading was counted up to once per SNP, even if that SNP was associated with multiple conditions corresponding to that subject heading. Using the same method and the same list of condition-subject heading associations, the number of times each subject heading appeared across the union set was also calculated. Additional counts were also generated for the subsets of pathogenic, benign, or unspecified SNPs in each set. Analysis was primarily carried out on the counts of top-level subject headings.

Additional figures were calculated for the percentage of diseases with at least one C > U SNP and/or editing site associated. For the purposes of this work, the third level of MeSH subject headings (grandchildren of top-level terms) were considered to represent individual diseases. For each top-level term and select second-level terms, the number of associated third-level terms was counted. If at least one C > U SNP or editing site mapped directly to a third-level term or to child terms of that third-level term, the term was counted as having a C > U SNP or editing site. The percentages of diseases with at least one associated C > U SNP and diseases with at least one editing site were then calculated by dividing these counts by the total number of third-level terms.

## Reporting summary

Further information on research design is available in the Nature Portfolio Reporting Summary linked to this article.

## Data availability

The data used in this work are publicly available. The RNA editing sites used to train and assess RNAsee were originally collated in Asaoka et al., and they may be accessed therein[9]. Information on known SNPs were collected from the ClinVar database, which can be accessed via https://www.ncbi.nlm.nih.gov/clinvar/. RNA coding sequences were sourced from the consensus coding sequence (CCDS) website, which may be accessed via https://www.ncbi.nlm.nih.gov/projects/CCDS. The CCDS files and ClinVar data used in this paper and the scores assigned to each site by RNAsee are available via http://compbio.buffalo.edu/data/mc_rnasee_biodiv/. The numerical source data behind the graphs can be found at http://compbio.buffalo.edu/data/mc_rnasee_biodiv/ as well. All other data were available from the corresponding author on reasonable request.

## Code availability

The RNAsee Python package is available on GitHub at https://github.com/ram-compbio/RNAsee and at http://compbio.buffalo.edu/software/rnasee. The exact version used here is accessible through https://doi.org/10.5281/zenodo.10892515[29].

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

## Acknowledgements

This work has been supported in part by grants from NIH NLM [T15LM012495 and 1R25LM014213], NIAAA [R21AA026954 and R33AA0226954], NIDA [1K01DA056690], NIST [60NANB22D168], and NCATS [UL1TR001412]. This study was funded in part by the Department of Veterans Affairs. The authors would like to acknowledge the Center for Computational Research (CCR) at the University at Buffalo for computational support. We would also like to thank all members of the Samudrala Computational Biology Group.

## Author contributions

M.V.N. updated RNAsee to version 2.0, conducted the benchmarking and biodiversity analyses presented herein, and drafted the manuscript. Z.F. developed the original version of RNAsee, assisted in developing RNAsee version 2.0, provided research design support, and edited the manuscript. S.M. assisted development of the original version of RNAsee. B.H.S. and B.E.B. provided critical insight into the functioning of APOBEC3A/G RNA editing, leading to the design of RNAsee. R.S. provided research design support, computational resources and expertise, edited the manuscript, and helped with project supervision. P.L.E. conceived and supervised the overall project, helped with approach and research design, provided expertise, and edited the manuscript. All authors provided feedback on the manuscript and have agreed to the published version.

## Competing interests

The authors declare no competing interests.
