## [Peer Review File · Communications Biology]

Reviewers' comments:

Reviewer #1 (Remarks to the Author):

The researchers used the RNAseq tool to explore how RNA editing, specifically by APOBEC3A/G enzymes converting cytosines to uracils, might reflect known DNA SNPs. They found that about 4.5% of non-synonymous SNPs leading to C>U changes in RNA, including 5.4% labeled as pathogenic, could be influenced by APOBEC3A/G editing. This suggests some proteins believed to arise from DNA mutations might also result from RNA editing, with potential health impacts. Notably, these edited SNPs were linked to certain diseases, emphasizing RNA editing's role in human health and disease.

However, I do have a couple of concerns about the study.

1. The authors have drawn a connection between SNPs in DNA from ClinVar and potential editing sites. However, there's ambiguity in understanding how RNA editing is reflected in DNA SNPs, given that ClinVar predominantly sources data not from RNA-seq experiments. This aspect remains a primary point of contention for me.
2. The current title overreaches in its scope. Given that the main focus of the research is on the potential disease implications of edited sites, it doesn't adequately address broader human biodiversity. I strongly recommend refining the title to provide a more accurate representation of the study's core focus.
3. While the abstract underscores the notion that RNA editing, potentially influenced by environmental factors, plays a substantial role in protein biodiversity leading to disease, the study does not offer conclusive evidence supporting the environmental stimulus of RNA editing. The emphasis on environmental factors as a catalyst for RNA editing seems overstated and could mislead readers regarding its significance.
4. Minor typos.

For example: page 16: . For each term in this list, the most relevant MeSH subject heading (descriptor or supplemental concept record) was found. MeSH mappings were used to "associated" supplemental concept records with tree terms unless a mapping contained a qualifier that suggested the supplemental concept record was not a child of the mapped term (such as "abnormalities").

Reviewer #2 (Remarks to the Author):

This manuscript discusses the role of C-to-U RNA editing in human biodiversity. It explores the significant contribution of RNA editing to our genetic diversity, expanding our understanding of the mechanisms behind intra-organism biodiversity. I have several critical comments for this manuscript.

1. Considering the data collection, have the authors considered removing samples with high sequence redundancy? Because these samples may lead to overfitting the model.
2. The authors should introduce the rules and algorithms used in the rules-based model in detail.
3. For the machine learning models, have the authors normalised the features used for model training?
4. For the imbalanced datasets, the authors should also report the performance under the Precision-Recall curves.

5. The authors are suggested to use model interpretation approaches to explain why the machine learning model works.
6. The authors should provide some examples in the GitHub repository to facilitate the users to use their method.
7. The authors should clarify why the random forest model performed better at the lower threshold. In addition, The legend of Figure 1B should be provided.
8. The authors should better present the corresponding F1 and MCC for the results of Figures 1C and 1D.
9. For Figure 2D, it is not clear why the number of the total is less than the number of exonic.
10. The authors should also provide an overall framework for three models they developed.

Response to Reviewers

Response to Reviewer 1

Comment 1: The authors have drawn a connection between SNPs in DNA from ClinVar and potential editing sites. However, there's ambiguity in understanding how RNA editing is reflected in DNA SNPs, given that ClinVar predominantly sources data not from RNA-seq experiments. This aspect remains a primary point of contention for me.

Response to Comment 1: Thank you for providing commentary on our article. It is true that a large portion of this article hinges on the acceptance of using DNA SNPs as a starting point for finding potential RNA editing sites. We can back up the validity of considering the sites we examined as potential RNA editing sites for two reasons. Firstly, APOBEC3A is a major driver of C>U deamination in DNA, and previous works suggest that APOBEC3A edits RNA at similar sites as DNA (see Buisson et al 2019 for DNA editing sites, Sharma et al 2017 for RNA editing sites, and Jalili et al 2020 for correspondence between DNA and RNA editing sites). We primarily examined those sites returned by RNAseq as potential editing sites. If RNAseq performed in application as it did in benchmarking, we would expect at least 250 (5.5%) of these sites to actually undergo RNA editing, resulting in the creation of these variant transcripts and proteins. Therefore, our results support the overlap of RNA editing with DNA mutations. Secondly, our examination of the DNA polymorphisms suggests that some level of C>U deamination is likely contributing to the population of DNA SNPs recorded in ClinVar; over 40% of SNPs are associated with a C>T change on one strand, as opposed to less than 20% of SNPs associated with the other form of deamination, A>I/A>G. **Therefore, it is reasonable to conclude that, though the ClinVar SNPs are primarily recorded from DNA, they include a population of mutations which result from APOBEC3A DNA editing and are thus, when transcribed, likely to also undergo APOBEC3-mediated RNA editing.**

As for connecting DNA SNPs and RNA editing, we believe that this is valid because we only considered those sites which are transcribed into RNA and, ultimately, translated into proteins. The effects on human health associated with these SNPs should therefore largely result from the creation of variant RNA and proteins. RNA editing can result in over 50% transcript alteration when a high-affinity sequence and activating environmental factors are involved (see Sharma et al 2015, Figure 1A for an example). We contend, therefore, that transient instances of high editing activity could cause transient creation of these same variant RNA and proteins. This, in turn, could result in similar dysfunctions on the cellular or tissue level as are observed in the DNA SNPs recorded in ClinVar. Of course, these effects would be less universal and more transient, but they could still have serious consequences if they occurred in the wrong tissues, such as the brain or heart, created the wrong type of variant protein, such as plaque-forming proteins or autoantigens, or occurred alongside a heterozygous DNA mutation affecting the same gene, altering the wild-type RNA and increasing the expression of a dosage-sensitive mutation such as *Pten* mutations (see Alimonti et al Nature Genetics, 2010 for an example of a dosage-sensitive mutation in *Pten*).

Because we anticipate readers will have similar concerns about the validity of drawing parallels between DNA SNPs and RNA editing, we have re-worked the discussion to include our reasoning, as presented above. We have also added cautionary statements regarding the translatability of information on DNA SNPs to RNA editing to the abstract and discussion.

***Comment 2:** The current title overreaches in its scope. Given that the main focus of the research is on the potential disease implications of edited sites, it doesn't adequately address broader human biodiversity. I strongly recommend refining the title to provide a more accurate representation of the study's core focus.*

Response to Comment 2: We have attempted to narrow the scope of the title to what is covered in this paper to just the disease implications of C>U editing. We are now proposing the following title: The Implications of APOBEC3-Mediated C-to-U RNA Editing for Human Disease.

***Comment 3:** While the abstract underscores the notion that RNA editing, potentially influenced by environmental factors, plays a substantial role in protein biodiversity leading to disease, the study does not offer conclusive evidence supporting the environmental stimulus of RNA editing. The emphasis on environmental factors as a catalyst for RNA editing seems overstated and could mislead readers regarding its significance.*

Response to Comment 3: Previous studies in 2013 and 2015 for APOBEC3A and in 2019 for APOBEC3G provide strong evidence that APOBEC3-mediated RNA editing is induced by interferons (APOBEC3A) and/or hypoxia (APOBEC3A and -3G), which are consequences of environmental stressors. The authors believe that environmental factors, for example, infections, which are known to stimulate interferon alpha are capable of activating APOBEC3A in doing so. This can lead to RNA editing and the subsequent effects on transcription and human disease. Thus, micro-environmental factors play an essential role in activating APOBEC3-mediated RNA editing. That being said, you are correct in stating that our study does not contribute substantial evidence in that area. Therefore, we have limited the discussion of environmental factors in the abstract, and we have instead added a sentence more to the introduction on previous research regarding environmental factors in RNA editing to inform readers without mis- or over-stating the findings of our study.

***Comment 4:** Minor typos. For example: page 16: For each term in this list, the most relevant MeSH subject heading (descriptor or supplemental concept record) was found. MeSH mappings were used to “associated” supplemental concept records with tree terms unless a mapping contained a qualifier that suggested the supplemental concept record was not a child of the mapped term (such as “abnormalities”).*

Response to Comment 4: Typos and other errors (e.g., sentence repetitions, word omissions) have been corrected throughout the manuscript. These changes are highlighted in red.

Response to Reviewer 2

Comment 1: Considering the data collection, have the authors considered removing samples with high sequence redundancy? Because these samples may lead to overfitting the model.

Response to Comment 1: Thank you for taking the time to provide commentary on our article. This is a good point. For this version of RNAsee, we did not take sequence redundancy into account. However, because the sites we trained on are known editing sites, we expect a certain amount of sequence redundancy due to common features of editing sites. That being said, we will consider this factor in future versions of RNAsee.

Comment 2: The authors should introduce the rules and algorithms used in the rules-based model in detail.

Response to Comment 2: The rules outlined in the RNAsee subsection of the Methods section has been clarified to include all rules on which potential editing sites are scored. A more thorough examination of all four models included in RNAsee will be presented in a separate technical paper that the authors are currently preparing for submission.

Comment 3: For the machine learning models, have the authors normalised the features used for model training?

Response to Comment 3: In this case, since the random forest model only considered binary features, which are naturally bounded between 0 and 1, we did not consider feature normalization to be a major concern.

Comment 4: For the imbalanced datasets, the authors should also report the performance under the Precision-Recall curves.

Response to Comment 4: We have re-worked the Results: Performance of RNAsee section to include a discussion of AUPRC in the same paragraph as the AUROC results are reported. We also added the precision-recall curve to Figure 1B.

Comment 5: The authors are suggested to use model interpretation approaches to explain why the machine learning model works.

Response to Comment 5: This is a very good point. For the purposes of this paper, we cover the feature vector of the model, which gives us an outer bound for this question. The features input into the training of the ML model were 50 bits representing the 15 nucleotides preceding and 10 nucleotides following the cytosine being examined, with each nucleotide being represented by 2 binary features: whether that nucleotide is a purine, and whether it is a G or C (ie can participate in GC pairing; see Methods: RNAsee). We also will address this in more detail in a future paper solely focused on RNAsee from a technical point of view. However, we are concerned that adding that content to this paper specifically might make it unwieldy, and we have thus attempted to pare down the content related to RNAsee's inner workings and development so that the methods can be broadly understood and their reliability adequately assessed by the reader.

Comment 6: The authors should provide some examples in the GitHub repository to facilitate the users to use their method.

Response to Comment 6: We have expanded the Github content to include an example of running each individual model from an interactive Python session, including the expected output on one of the test files

(corresponding to the SDHB gene, in which APOBEC3A-mediated RNA editing was first characterized at cytosine 136).

Comment 7: The authors should clarify why the random forest model performed better at the lower threshold. In addition, the legend of Figure 1B should be provided.

Response to Comment 7: The rules-based model's worse performance at low thresholds can largely be attributed to its inclusion criteria for cytosines, which excludes some known editing sites and caps its recall at the lowest score threshold (-1). This clarification has been added to the re-worked AUROC/AUPRC paragraph. Figure 1B has also been given a separate legend, which also includes the baseline that was added along with the precision-recall curve for clarity.

Comment 8: The authors should better present the corresponding F1 and MCC for the results of Figures 1C and 1D.

Response to Comment 8: Due to space limitations, these metrics were not added to Figure 1, but they were calculated and incorporated into the Results: Performance of RNAseq section on both the testing and proportional sets. Because this addition lengthened this section, the section was re-organized for easier reading and more clarity as to which metric result corresponded to which dataset.

Comment 9: For Figure 2D, it is not clear why the number of the total is less than the number of exonic.

Response to Comment 9: In Figure 2D, since only exonic SNPs were extracted and analyzed from ClinVar, "total exonic" refers to the total number of SNPs extracted. Then, "total C>U SNPs" refers to the subset of exonic SNPs extracted which resulted in a C>U change in RNA. We have updated the wording in Figure 2D and elsewhere to simply "C>U SNPs" to make the terminology clearer and more consistent.

Comment 10: The authors should also provide an overall framework for three models they developed.

Response to Comment 10: We have attempted to explain the data used in and processes used by the four models, particularly the rules of the rules-based model and the training process of the random forest model, sufficiently yet succinctly in the Methods section. We will more completely address the reasoning behind the creation and components of our models in the technical paper, but we are concerned that adding that content to this paper will make it unwieldy.

Reviewers' comments:

Reviewer #1 (Remarks to the Author):

The authors have well addressed my concerns in this version. I don't have further comments.

Reviewer #1 on the response to reviewer #2

The authors have satisfactorily addressed the majority of the comments. However, I recommend that the authors evaluate the level of sequence redundancy within the sequences used in their analysis, and the authors could just post in the response letter. A moderate level of redundancy would not pose a significant concern, but I would like to verify the status of the dataset employed.

Response to the Reviewer comments:

We examined the sequence identity of the 25 nucleotide sequences surrounding each known editing site (positive sample), including 15 nucleotides preceding and 10 nucleotides following the edited C. These sequences comprise the dataset used to train and test the machine learning algorithm. The edited C was not considered for these calculations, as it is necessarily always cytosine. For each of the 4,760,155 unique sequence-sequence pairs, which do not include self-matches or repeated pairings, we measured the sequence identity, or number of identical nucleotides, between the two sequences. We also repeated this process for the negative sample, which included 440,418 sites (96,983,787,153 unique sequence-sequence pairs).

With regard to the positive samples, 13 pairs (<0.01% of all pairs) had over 90% sequence identity (at least 23 out of 25 nucleotides in common), 123 (<0.01%) had over 75% sequence identity (at least 19 nucleotides), and 130518 (2.74%) had over 50% (at least 13 nucleotides) sequence identity. According to “An Introduction to Sequence Similarity (“Homology”) Searching” by Pearson WR (2013), 30% sequence identity is a standard threshold for determining whether two sequences are related; 2411459 (50.66%) pairs had over 30% sequence identity (at least 8 nucleotides). Sequence identity was lower in the negative samples, with <0.01% above 90% and 75% sequence identity, 0.53% above 50%, and 28.8% above 30%.

Overall, we would expect some level of sequence similarity among positive sequences given they are all edited by APOBEC3A/G enzymes, the activity of which is specific to certain sequential and structural features. Since approximately half of positive and more than 70% of negative pairings had a sequence identity lower than the 30% identity threshold, there appears to be a low to moderate level of sequence redundancy. Therefore, we do not expect that sequence redundancy in the training dataset is significantly biasing the model.

REVIEWERS' COMMENTS:

Reviewer #1 (Remarks to the Author):

Given the context of redundancy, sequence similarity presents a concern. However, since there are significantly fewer positive samples compared to negative ones, this issue may be less impactful in this scenario. Despite this, it is advisable for the authors to remove duplicates while constructing the datasets before they are split into training and testing sets in future endeavors.